# NeRV: Neural Representations for Videos

**Hao Chen**[1], **Bo He**[1], **Hanyu Wang**[1], **Yixuan Ren**[1], **Ser-Nam Lim**[2], **Abhinav Shrivastava**[1]

[1]University of Maryland, College Park, [2]Facebook AI

{chenh, bohe, hywang66, yxren, abhinav}@umd.edu, sernamlim@fb.com

## Abstract

We propose a novel neural representation for videos (NeRV) which encodes videos in neural networks. Unlike conventional representations that treat videos as frame sequences, we represent videos as neural networks taking frame index as input. Given a frame index, NeRV outputs the corresponding RGB image. Video encoding in NeRV is simply fitting a neural network to video frames and decoding process is a simple feedforward operation. As an image-wise implicit representation, NeRV output the whole image and shows great efficiency compared to pixel-wise implicit representation, improving the encoding speed by **25**× to **70**×, the decoding speed by **38**× to **132**×, while achieving better video quality. With such a representation, we can treat videos as neural networks, simplifying several video-related tasks. For example, conventional video compression methods are restricted by a long and complex pipeline, specifically designed for the task. In contrast, with NeRV, we can use any neural network compression method as a proxy for video compression, and achieve comparable performance to traditional frame-based video compression approaches (H.264, HEVC *etc*.). Besides compression, we demonstrate the generalization of NeRV for video denoising. The source code and pre-trained model can be found at https://github.com/haochen-rye/NeRV.git.

## 1 Introduction

What is a video? Typically, a video captures a dynamic visual scene using a sequence of frames. A schematic interpretation of this is a curve in 2D space, where each point can be characterized with a $(x, y)$ pair representing the spatial state. If we have a model for all $(x, y)$ pairs, then, given any $x$, we can easily find the corresponding $y$ state. Similarly, we can interpret a video as a recording of the visual world, where we can find a corresponding RGB state for every single timestamp. This leads to our main claim: *can we represent a video as a function of time?*

More formally, can we represent a video $V$ as $V = \{v_t\}_{t=1}^T$, where $v_t = f_\theta(t)$, *i.e.*, a frame at timestamp $t$, is represented as a function $f$ parameterized by $\theta$. Given their remarkable representational capacity [1], we choose deep neural networks as the function in our work. Given these intuitions, we propose NeRV, a novel representation that represents videos as implicit functions and encodes them into neural networks. Specifically, with a fairly simple deep neural network design, NeRV can reconstruct the corresponding video frames with high quality, given the frame index. Once the video is encoded into a neural network, this network can be used as a proxy for video, where we can directly extract all video information from the representation. Therefore, unlike traditional video representations which treat videos as sequences of frames, shown in Figure 1 (a), our proposed NeRV considers a video as a unified neural network with all information embedded within its architecture and parameters, shown in Figure 1 (b).

As an image-wise implicit representation, NeRV shares lots of similarities with pixel-wise implicit visual representations [5, 6] which takes spatial-temporal coordinates as inputs. The main differences between our work and image-wise implicit representation are the output space and architecture

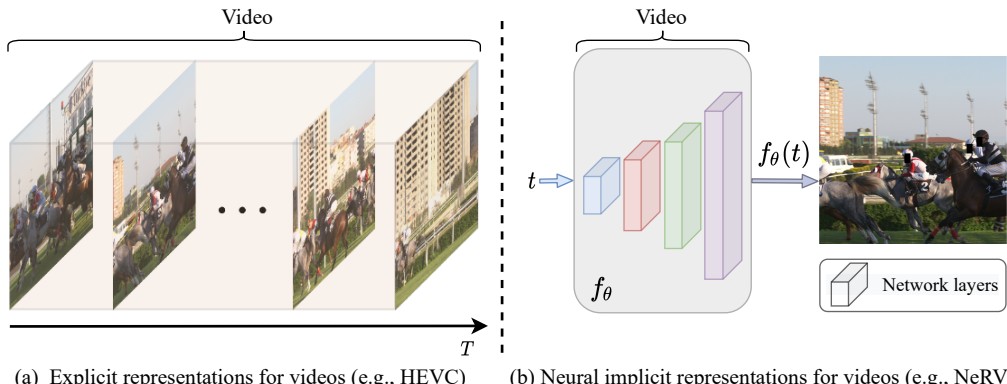

(a) Explicit representations for videos (e.g., HEVC)  (b) Neural implicit representations for videos (e.g., NeRV)

Figure 1: **(a)** Conventional video representation as **frame sequences**. **(b)** NeRV, representing video as **neural networks**, which consists of multiple convolutional layers, taking the normalized frame index as the input and output the corresponding RGB frame.

Table 1: Comparison of different video representations. Although explicit representations outperform implicit ones in encoding speed and compression ratio now, NeRV shows great advantage in decoding speed. And NeRV outperforms pixel-wise implicit representations in all metrics.

|  | Explicit (frame-based) | | Implicit (unified) | |
| --- | --- | --- | --- | --- |
|  | Hand-crafted (*e.g.*, HEVC [2]) | Learning-based (*e.g.*, DVC [3]) | Pixel-wise (*e.g.*, NeRF [4]) | Image-wise (Ours) |
| Encoding speed | **Fast** | Medium | Very slow | Slow |
| Decoding speed | Medium | Slow | Very slow | **Fast** |
| Compression ratio | Medium | **High** | Low | Medium |

designs. Pixel-wise representations output the RGB value for each pixel, while NeRV outputs a whole image, demonstrated in Figure 2. Given a video with size of $T \times H \times W$, pixel-wise representations need to sample the video $T * H * W$ times while NeRV only need to sample $T$ times. Considering the huge pixel number, especially for high resolution videos, NeRV shows great advantage for both encoding time and decoding speed. Different output space also leads to different architecture designs, NeRV utilizes a MLP + ConvNets architecture to output an image while pixel-wise representation uses a simple MLP to output the RGB value of the pixel. Sampling efficiency of NeRV also simplify the optimization problem, which leads to better reconstruction quality compared to pixel-wise representations.

We also demonstrate the flexibility of NeRV by exploring several applications it affords. Most notably, we examine the suitability of NeRV for video compression. Traditional video compression frameworks are quite involved, such as specifying key frames and inter frames, estimating the residual information, block-size the video frames, applying discrete cosine transform on the resulting image blocks and so on. Such a long pipeline makes the decoding process very complex as well. In contrast, given a neural network that encodes a video in NeRV, we can simply cast the video compression task as a model compression problem, and trivially leverage any well-established or cutting edge model compression algorithm to achieve good compression ratios. Specifically, we explore a three-step model compression pipeline: model pruning, model quantization, and weight encoding, and show the contributions of each step for the compression task. We conduct extensive experiments on popular video compression datasets, such as UVG [7], and show the applicability of model compression techniques on NeRV for video compression. We briefly compare different video representations in Table 1 and NeRV shows great advantage in decoding speed.

Besides video compression, we also explore other applications of the NeRV representation for the video denoising task. Since NeRV is a learnt implicit function, we can demonstrate its robustness to noise and perturbations. Given a noisy video as input, NeRV generates a high-quality denoised output, without any additional operation, and even outperforms conventional denoising methods.

The contribution of this paper can be summarized into four parts:

- We propose NeRV, a novel image-wise implicit representation for videos, representating a video as a neural network, converting video encoding to model fitting and video decoding as a simple feedforward operation.

- Compared to pixel-wise implicit representation, NeRV output the whole image and shows great efficiency, improving the encoding speed by $25\times$ to $70\times$, the decoding speed by $38\times$ to $132\times$, while achieving better video quality.

- NeRV allows us to convert the video compression problem to a model compression problem, allowing us to leverage standard model compression tools and reach comparable performance with conventional video compression methods, *e.g.*, H.264 [8], and HEVC [2].

- As a general representation for videos, NeRV also shows promising results in other tasks, *e.g.*, video denoising. Without any special denoisng design, NeRV outperforms traditional hand-crafted denoising algorithms (medium filter *etc*.) and ConvNets-based denoisng methods.

## 2   Related Work

**Implicit Neural Representation.** Implicit neural representation is a novel way to parameterize a variety of signals. The key idea is to represent an object as a function approximated via a neural network, which maps the coordinate to its corresponding value (*e.g.*, pixel coordinate for an image and RGB value of the pixel). It has been widely applied in many 3D vision tasks, such as 3D shapes [9, 10], 3D scenes [11–14], and appearance of the 3D structure [4, 15, 16]. Comparing to explicit 3D representations, such as voxel, point cloud, and mesh, the continuous implicit neural representation can compactly encode high-resolution signals in a memory-efficient way. Most recently, [17] demonstrated the feasibility of using implicit neural representation for image compression tasks. Although it is not yet competitive with the state-of-the-art compression methods, it shows promising and attractive proprieties. In previous methods, MLPs are often used to approximate the implicit neural representations, which take the spatial or spatio-temporal coordinate as the input and output the signals at that single point (*e.g.*, RGB value, volume density). In contrast, our NeRV representation, trains a purposefully designed neural network composed of MLPs and convolution layers, and takes the frame index as input and directly outputs all the RGB values of that frame.

**Video Compression.** As a fundamental task of computer vision and image processing, visual data compression has been studied for several decades. Before the resurgence of deep networks, handcrafted image compression techniques, like JPEG [18] and JPEG2000 [19], were widely used. Building upon them, many traditional video compression algorithms, such as MPEG [20], H.264 [8], and HEVC [2], have achieved great success. These methods are generally based on transform coding like Discrete Cosine Transform (DCT) [21] or wavelet transform [22], which are well-engineered and tuned to be fast and efficient. More recently, deep learning-based visual compression approaches have been gaining popularity. For video compression, the most common practice is to utilize neural networks for certain components while using the traditional video compression pipeline. For example, [23] proposed an effective image compression approach and generalized it into video compression by adding interpolation loop modules. Similarly, [24] converted the video compression problem into an image interpolation problem and proposed an interpolation network, resulting in competitive compression quality. Furthermore, [25] generalized optical flow to scale-space flow to better model uncertainty in compression. Later, [26] employed a temporal hierarchical structure, and trained neural networks for most components including key frame compression, motion estimation, motions compression, and residual compression. However, all of these works still follow the overall pipeline of traditional compression, arguably limiting their capabilities.

**Model Compression.** The goal of model compression is to simplify an original model by reducing the number of parameters while maintaining its accuracy. Current research on model compression research can be divided into four groups: parameter pruning and quantization [27–32]; low-rank factorization [33–35]; transferred and compact convolutional filters [36–39]; and knowledge distillation [40–43]. Our proposed NeRV enables us to reformulate the video compression problem into model compression, and utilize standard model compression techniques. Specifically, we use model pruning and quantization to reduce the model size without significantly deteriorating the performance.

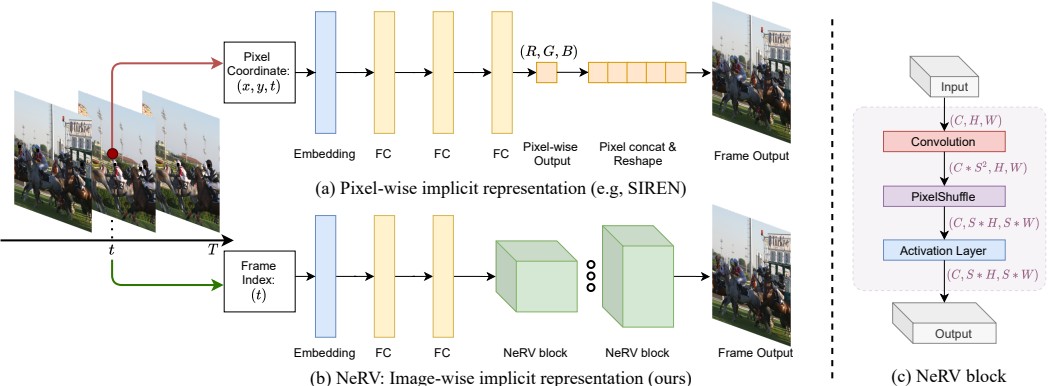

Figure 2: **(a) Pixel-wise** implicit representation taking pixel coordinates as input and use a simple MLP to output pixel RGB value **(b) NeRV: Image-wise** implicit representation taking frame index as input and use a MLP + ConvNets to output the whole image. **(c) NeRV block** architecture, upscale the feature map by $S$ here.

# 3 Neural Representations for Videos

We first present the NeRV representation in Section 3.1, including the input embedding, the network architecture, and the loss objective. Then, we present model compression techniques on NeRV in Section 3.2 for video compression.

## 3.1 NeRV Architecture

In NeRV, each video $V = \{v_t\}_{t=1}^T \in \mathbb{R}^{T \times H \times W \times 3}$ is represented by a function $f_\theta : \mathbb{R} \to \mathbb{R}^{H \times W \times 3}$, where the input is a frame index $t$ and the output is the corresponding RGB image $v_t \in \mathbb{R}^{H \times W \times 3}$. The encoding function is parameterized with a deep neural network $\theta$, $v_t = f_\theta(t)$. Therefore, video encoding is done by fitting a neural network $f_\theta$ to a given video, such that it can map each input timestamp to the corresponding RGB frame.

**Input Embedding.** Although deep neural networks can be used as universal function approximators [1], directly training the network $f_\theta$ with input timestamp $t$ results in poor results, which is also observed by [4, 44]. By mapping the inputs to a high embedding space, the neural network can better fit data with high-frequency variations. Specifically, in NeRV, we use Positional Encoding [4, 6, 45] as our embedding function

$$\Gamma(t) = \left(\sin\left(b^0\pi t\right), \cos\left(b^0\pi t\right), \ldots, \sin\left(b^{l-1}\pi t\right), \cos\left(b^{l-1}\pi t\right)\right) \tag{1}$$

where $b$ and $l$ are hyper-parameters of the networks. Given an input timestamp $t$, normalized between $(0, 1]$, the output of embedding function $\Gamma(\cdot)$ is then fed to the following neural network.

**Network Architecture.** NeRV architecture is illustrated in Figure 2 (b). NeRV takes the time embedding as input and outputs the corresponding RGB Frame. Leveraging MLPs to directly output all pixel values of the frames can lead to huge parameters, especially when the images resolutions are large. Therefore, we stack multiple NeRV blocks following the MLP layers so that pixels at different locations can share convolutional kernels, leading to an efficient and effective network. Inspired by the super-resolution networks, we design the NeRV block, illustrated in Figure 2 (c), adopting PixelShuffle technique [46] for upscaling method. Convolution and activation layers are also inserted to enhance the expressibilty. The detailed architecture can be found in the supplementary material.

**Loss Objective.** For NeRV, we adopt combination of L1 and SSIM loss as our loss function for network optimization, which calculates the loss over all pixel locations of the predicted image and the ground-truth image as following

$$L = \frac{1}{T} \sum_{t=1}^T \alpha \left\| f_\theta(t) - v_t \right\|_1 + (1 - \alpha)(1 - \text{SSIM}(f_\theta(t), v_t)) \tag{2}$$

where $T$ is the frame number, $f_\theta(t) \in \mathbb{R}^{H \times W \times 3}$ the NeRV prediction, $v_t \in \mathbb{R}^{H \times W \times 3}$ the frame ground truth, $\alpha$ is hyper-parameter to balance the weight for each loss component.

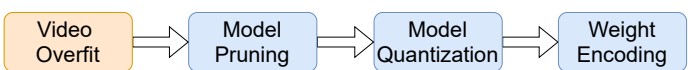

Figure 3: NeRV-based video compression pipeline.

## 3.2 Model Compression

In this section, we briefly revisit model compression techniques used for video compression with NeRV. Our model compression composes of four standard sequential steps: video overfit, model pruning, weight quantization, and weight encoding as shown in Figure 3.

**Model Pruning.** Given a neural network fit on a video, we use global unstructured pruning to reduce the model size first. Based on the magnitude of weight values, we set weights below a threshold as zero,

$$\theta_i = \begin{cases} \theta_i, & \text{if } \theta_i \geq \theta_q \\ 0, & \text{otherwise,} \end{cases} \tag{3}$$

where $\theta_q$ is the $q$ percentile value for all parameters in $\theta$. As a normal practice, we fine-tune the model to regain the representation, after the pruning operation.

**Model Quantization.** After model pruning, we apply model quantization to all network parameters. Note that different from many recent works [31, 47–49] that utilize quantization during training, NeRV is only quantized post-hoc (after the training process). Given a parameter tensor $\mu$

$$\mu_i = \text{round}\left(\frac{\mu_i - \mu_{\min}}{2^{\text{bit}}}\right) * \text{scale} + \mu_{\min}, \quad \text{scale} = \frac{\mu_{\max} - \mu_{\min}}{2^{\text{bit}}} \tag{4}$$

where 'round' is rounding value to the closest integer, 'bit' the bit length for quantized model, $\mu_{\max}$ and $\mu_{\min}$ the max and min value for the parameter tensor $\mu$, 'scale' the scaling factor. Through Equation 4, each parameter can be mapped to a 'bit' length value. The overhead to store 'scale' and $\mu_{\min}$ can be ignored given the large parameter number of $\mu$, e.g., they account for only $0.005\%$ in a small $3 \times 3$ Conv with $64$ input channels and $64$ output channels ($37k$ parameters in total).

**Entropy Encoding.** Finally, we use entropy encoding to further compress the model size. By taking advantage of character frequency, entropy encoding can represent the data with a more efficient codec. Specifically, we employ Huffman Coding [50] after model quantization. Since Huffman Coding is lossless, it is guaranteed that a decent compression can be achieved without any impact on the reconstruction quality. Empirically, this further reduces the model size by around 10%.

## 4 Experiments

### 4.1 Datasets and Implementation Details

We perform experiments on "Big Buck Bunny" sequence from scikit-video to compare our NeRV with pixel-wise implicit representations, which has 132 frames of $720 \times 1080$ resolution. To compare with state-of-the-arts methods on video compression task, we do experiments on the widely used UVG [7], consisting of 7 videos and 3900 frames with $1920 \times 1080$ in total.

In our experiments, we train the network using Adam optimizer [51] with learning rate of 5e-4. For ablation study on UVG, we use cosine annealing learning rate schedule [52], batchsize of 1, training epochs of 150, and warmup epochs of 30 unless otherwise denoted. When compare with state-of-the-arts, we run the model for 1500 epochs, with batchsize of 6. For experiments on "Big Buck Bunny", we train NeRV for 1200 epochs unless otherwise denoted. For fine-tune process after pruning, we use 50 epochs for both UVG and "Big Buck Bunny".

For NeRV architecture, there are 5 NeRV blocks, with up-scale factor 5, 3, 2, 2, 2 respectively for 1080p videos, and 5, 2, 2, 2, 2 respectively for 720p videos. By changing the hidden dimension of MLP and channel dimension of NeRV blocks, we can build NeRV model with different sizes. For input embedding in Equation 1, we use $b = 1.25$ and $l = 80$ as our default setting. For loss objective in Equation 2, $\alpha$ is set to 0.7. We evaluate the video quality with two metrics: PSNR and MS-SSIM [53]. Bits-per-pixel (BPP) is adopted to indicate the compression ratio. We implement our model in PyTorch [54] and train it in full precision (FP32). All experiments are run with NVIDIA RTX2080ti. Please refer to the supplementary material for more experimental details, results, and visualizations (*e.g.*, MCL-JCV [55] results)

Table 2: Compare with pixel-wise implicit representations. Training speed means time/epoch, while encoding time is the total training time.

| Methods | Parameters | Training Speed ↑ | Encoding Time ↓ | PSNR ↑ | Decoding FPS ↑ |
|---|---|---|---|---|---|
| SIREN [5] | 3.2M | 1× | 2.5× | 31.39 | 1.4 |
| NeRF [4] | 3.2M | 1× | 2.5× | 33.31 | 1.4 |
| NeRV-S (ours) | 3.2M | **25×** | **1×** | **34.21** | **54.5** |
| SIREN [5] | 6.4M | 1× | 5× | 31.37 | 0.8 |
| NeRF [4] | 6.4M | 1× | 5× | 35.17 | 0.8 |
| NeRV-M (ours) | 6.3M | **50×** | **1×** | **38.14** | **53.8** |
| SIREN [5] | 12.7M | 1× | 7× | 25.06 | 0.4 |
| NeRF [4] | 12.7M | 1× | 7× | 37.94 | 0.4 |
| NeRV-L (ours) | 12.5M | **70×** | **1×** | **41.29** | **52.9** |

Table 3: PSNR *vs.* epochs. Since video encoding of NeRV is an over-fit process, the reconstructed video quality keeps increasing with more training epochs. NeRV-S/M/L mean models with different sizes.

| Epoch | NeRV-S | NeRV-M | NeRV-L |
|---|---|---|---|
| 300 | 32.21 | 36.05 | 39.75 |
| 600 | 33.56 | 37.47 | 40.84 |
| 1.2k | 34.21 | 38.14 | 41.29 |
| 1.8k | 34.33 | 38.32 | 41.68 |
| 2.4k | **34.86** | **38.7** | **41.99** |

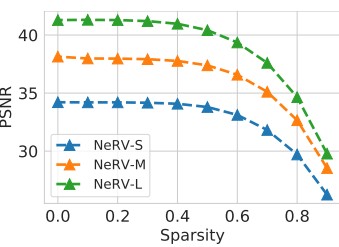

Figure 4: Model **pruning**. Sparsity is the ratio of parameters pruned.

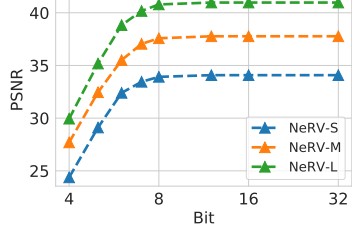

Figure 5: Model **quantization**. Bit is the bit length used to represent parameter value.

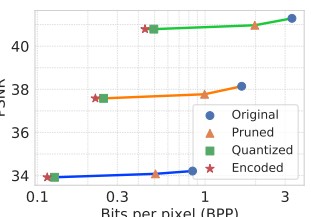

Figure 6: Compression **pipeline** to show how much each step contribute to compression ratio.

## 4.2 Main Results

We compare NeRV with pixel-wise implicit representations on 'Big Buck Bunny' video. We take SIREN [5] and NeRF [4] as the baseline, where SIREN [5] takes the original pixel coordinates as input and uses $sine$ activations, while NeRF [4] adds one positional embedding layer to encode the pixel coordinates and uses ReLU activations. Both SIREN and FFN use a 3-layer perceptron and we change the hidden dimension to build model of different sizes. For fair comparison, we train SIREN and FFN for 120 epochs to make encoding time comparable. And we change the filter width to build NeRV model of comparable sizes, named as NeRV-S, NeRV-M, and NeRV-L. In Table 2, NeRV outperforms them greatly in both encoding speed, decoding quality, and decoding speed. Note that NeRV can improve the training speed by $25\times$ to $70\times$, and speedup the decoding FPS by $38\times$ to $132\times$. We also conduct experiments with different training epochs in Table 3, which clearly shows that longer training time can lead to much better overfit results of the video and we notice that the final performances have not saturated as long as it trains for more epochs.

## 4.3 Video Compression

**Compression ablation.** We first conduct ablation study on video "Big Buck Bunny". Figure 4 shows the results of different pruning ratios, where model of 40% sparsity still reach comparable performance with the full model. As for model quantization step in Figure 5, a 8-bit model still remains the video quality compared to the original one (32-bit). Figure 6 shows the full compression pipeline with NeRV. The compression performance is quite robust to NeRV models of different sizes, and each step shows consistent contribution to our final results. Please note that we only explore these three common compression techniques here, and we believe that other well-established and cutting edge model compression algorithm can be applied to further improve the final performances of video compression task, which is left for future research.

**Compare with state-of-the-arts methods.** We then compare with state-of-the-arts methods on UVG dataset. First, we concatenate 7 videos into one single video along the time dimension and train NeRV on all the frames from different videos, which we found to be more beneficial than training a single model for each video. After training the network, we apply model pruning, quantization, and weight encoding as described in Section 3.2. Figure 7 and Figure 8 show the rate-distortion curves. We compare with H.264 [8], HEVC [2], STAT-SSF-SP [56], HLVC [26], Scale-space [25], and Wu *et al.* [24]. H.264 and HEVC are performed with *medium* preset mode. As the first image-wise neural

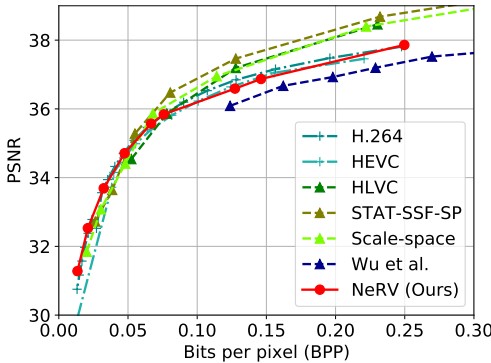
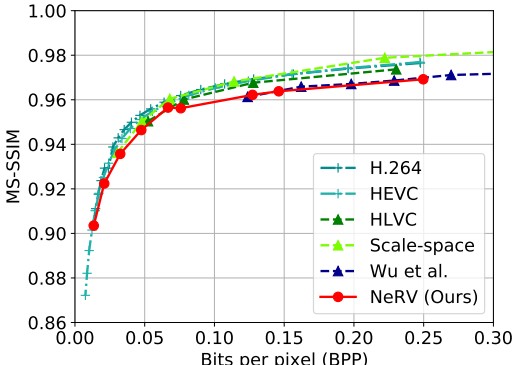

Figure 7: PSNR *vs*. BPP on UVG dataset.

Figure 8: MS-SSIM *vs*. BPP on UVG dataset.

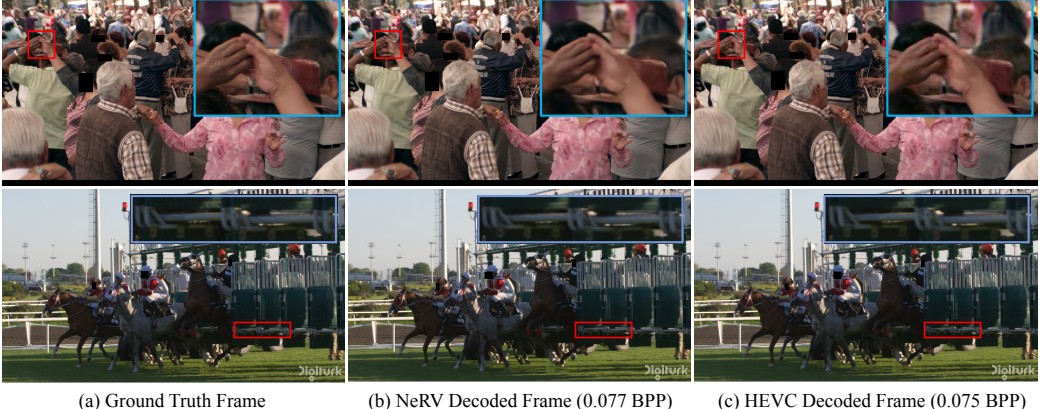

| (a) Ground Truth Frame | (b) NeRV Decoded Frame (0.077 BPP) | (c) HEVC Decoded Frame (0.075 BPP) |

Figure 9: Video compression visualization. At similar BPP, NeRV reconstructs videos with better details.

representation, NeRV generally achieves comparable performance with traditional video compression techniques and other learning-based video compression approaches. It is worth noting that when BPP is small, NeRV can match the performance of the state-of-the-art method, showing its great potential in high-rate video compression. When BPP becomes large, the performance gap is mostly because of the lack of full training due to GPU resources limitations. As shown in Table 3, the decoding video quality keeps increasing when the training epochs are longer. Figure 9 shows visualizations for decoding frames. At similar memory budget, NeRV shows image details with better quality.

**Decoding time** We compare with other methods for decoding time under a similar memory budget. Note that HEVC is run on CPU, while all other learning-based methods are run on a single GPU, including our NeRV. We speedup NeRV by running it in half precision (FP16). Due to the simple decoding process (feedforward operation), NeRV shows great advantage, even for carefully-optimized H.264. And lots of speepup can be expected by running quantizaed model on special hardware. All the other video compression methods have two types of frames: key and interval frames. Key frame can be reconstructed by its encoded feature only while the interval frame reconstruction is also based on the reconstructed key frames. Since most video frames are interval frames, their decoding needs to be done in a sequential manner after the reconstruction of the respective key frames. On the contrary, our NeRV can output frames at any random time index independently, thus making parallel decoding much simpler. This can be viewed as a distinct advantage over other methods.

## 4.4 Video Denoising

We apply several common noise patterns on the original video and train the model on the perturbed ones. During training, no masks or noise locations are provided to the model, *i.e.*, the target of the model is the noisy frames while the model has no extra signal of whether the input is noisy or not. Surprisingly, our model tries to avoid the influence of the noise and regularizes them implicitly with

Table 4: **Decoding speed** with BPP 0.2 for 1080p videos

| Methods | FPS ↑ |
|---|---|
| Habibian et al. [14] | $10^{-3.7}$ |
| Wu et al. [24] | $10^{-3}$ |
| Rippel et al. [57] | 1 |
| DVC [3] | 1.8 |
| Liu et al [58] | 3 |
| H.264 [8] | 9.2 |
| NeRV (FP32) | 5.6 |
| NeRV (FP16) | **12.5** |

Table 5: PSNR results for **video denoising**. "baseline" refers to the noisy frames before any denoising

| noise | white ↑ | black ↑ | salt & pepper ↑ | random ↑ | Average ↑ |
|---|---|---|---|---|---|
| Baseline | 27.85 | 28.29 | 27.95 | 30.95 | 28.74 |
| Gaussian | 30.27 | 30.14 | 30.23 | 30.99 | 30.41 |
| Uniform | 29.11 | 29.06 | 29.10 | 29.63 | 29.22 |
| Median | **33.89** | 33.84 | 33.87 | 33.89 | 33.87 |
| Minimum | 20.55 | 16.60 | 18.09 | 18.20 | 18.36 |
| Maximum | 16.16 | 20.26 | 17.69 | 17.83 | 17.99 |
| NeRV | 33.31 | **34.20** | **34.17** | **34.80** | **34.12** |

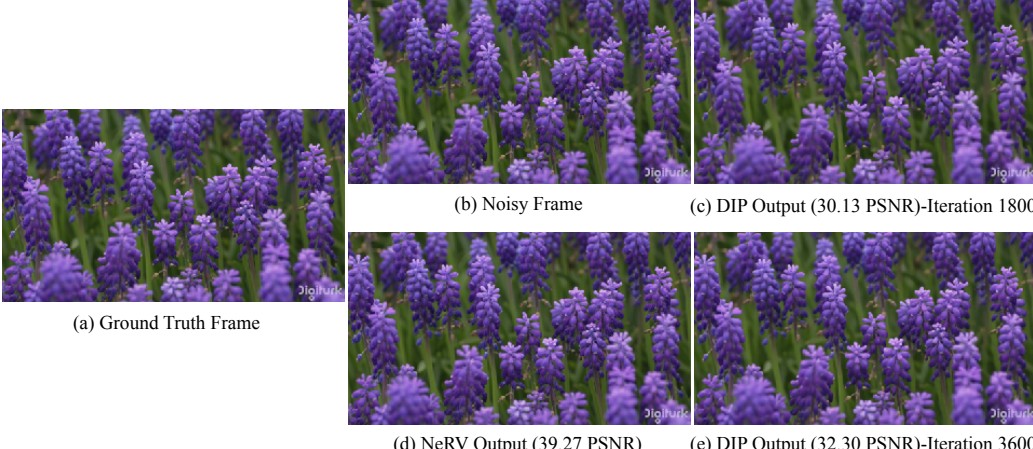

(a) Ground Truth Frame

(b) Noisy Frame

(c) DIP Output (30.13 PSNR)-Iteration 1800

(d) NeRV Output (39.27 PSNR)

(e) DIP Output (32.30 PSNR)-Iteration 3600

Figure 10: Denoising visualization. (c) and (e) are denoising output for DIP [59]. Data generalization of NeRV leads to robust and better denoising performance since all frames share the same representation, while DIP model overfits one model to one image only.

little harm to the compression task simultaneously, which can serve well for most partially distorted videos in practice.

The results are compared with some standard denoising methods including Gaussian, uniform, and median filtering. These can be viewed as denoising upper bound for any additional compression process. As listed in Table 5, the PSNR of NeRV output is usually much higher than the noisy frames although it's trained on the noisy target in a fully supervised manner, and has reached an acceptable level for general denoising purpose. Specifically, median filtering has the best performance among the traditional denoising techniques, while NeRV outperforms it in most cases or is at least comparable without any extra denoising design in both architecture design and training strategy.

We also compare NeRV with another neural-network-based denoising method, Deep Image Prior (DIP) [59]. Although its main target is image denoising, NeRV outperforms it in both qualitative and quantitative metrics, demonstrated in Figure 10. The main difference between them is that denoising of DIP only comes form architecture prior, while the denoising ability of NeRV comes from both architecture prior and data prior. DIP emphasizes that its image prior is only captured by the network structure of Convolution operations because it only feeds on a single image. But the training data of NeRV contain many video frames, sharing lots of visual contents and consistences. As a result, image prior is captured by both the network structure and the training data statistics for NeRV. DIP relies significantly on a good early stopping strategy to prevent it from overfitting to the noise. Without the noise prior, it has to be used with fixed iterations settings, which is not easy to generalize to any random kind of noises as mentioned above. By contrast, NeRV is able to handle this naturally by keeping training because the full set of consecutive video frames provides a strong regularization on image content over noise.

Table 6: Input embedding ablation. PE means positional encoding

| | PSNR | MS-SSIM |
|---|---|---|
| None | 24.93 | 0.769 |
| PE | **37.26** | **0.970** |

Table 7: Upscale layer ablation

| | PSNR | MS-SSIM |
|---|---|---|
| Bilinear Pooling | 29.56 | 0.873 |
| Transpose Conv | 36.63 | 0.967 |
| PixelShuffle | **37.26** | **0.970** |

Table 8: Norm layer ablation

| | PSNR | MS-SSIM |
|---|---|---|
| BatchNorm | 36.71 | **0.971** |
| InstanceNorm | 35.5 | 0.963 |
| None | **37.26** | 0.970 |

Table 9: Activation function ablation

| | PSNR | MS-SSIM |
|---|---|---|
| ReLU | 35.89 | 0.963 |
| Leaky ReLU | 36.76 | 0.968 |
| Swish | 37.08 | 0.969 |
| GELU | **37.26** | **0.970** |

Table 10: Loss objective ablation

| L2 | L1 | SSIM | PSNR | MS-SSIM |
|---|---|---|---|---|
| ✓ | | | 35.64 | 0.956 |
| | ✓ | | 35.77 | 0.959 |
| | | ✓ | 35.69 | **0.971** |
| ✓ | ✓ | | 35.95 | 0.960 |
| ✓ | | ✓ | 36.46 | 0.970 |
| | ✓ | ✓ | **37.26** | 0.970 |

## 4.5 Ablation Studies

Finally, we provide ablation studies on the UVG dataset. PSNR and MS-SSIM are adopted for evaluation of the reconstructed videos.

**Input embedding.** In Table 6, PE means positional encoding as in Equation 1, which greatly improves the baseline, None means taking the frame index as input directly. Similar findings can be found in [4], without any input embedding, the model can not learn high-frequency information, resulting in much lower performance.

**Upscale layer.** In Table 7, we show results of three different upscale methods. *i.e.*, Bilinear Pooling, Transpose Convolution, and PixelShuffle [46]. With similar model sizes, PixelShuffle shows best results. Please note that although Transpose convolution [60] reach comparable results, it greatly slowdown the training speed compared to the PixelShuffle.

**Normalization layer.** In Table 8, we apply common normalization layers in NeRV block. The default setup, without normalization layer, reaches the best performance and runs slightly faster. We hypothesize that the normalization layer reduces the over-fitting capability of the neural network, which is contradictory to our training objective.

**Activation layer.** Table 9 shows results for common activation layers. The GELU [61] activation function achieve the highest performances, which is adopted as our default design.

**Loss objective.** We show loss objective ablation in Table 10. We shows performance results of different combinations of L2, L1, and SSIM loss. Although adopting SSIM alone can produce the highest MS-SSIM score, but the combination of L1 loss and SSIM loss can achieve the best trade-off between the PSNR performance and MS-SSIM score.

## 5 Discussion

**Conclusion.** In this work, we present a novel neural representation for videos, NeRV, which encodes videos into neural networks. Our key sight is that by directly training a neural network with video frame index and output corresponding RGB image, we can use the weights of the model to represent the videos, which is totally different from conventional representations that treat videos as consecutive frame sequences. With such a representation, we show that by simply applying general model compression techniques, NeRV can match the performances of traditional video compression approaches for the video compression task, without the need to design a long and complex pipeline. We also show that NeRV can outperform standard denoising methods. We hope that this paper can inspire further research works to design novel class of methods for video representations.

**Limitations and Future Work.** There are some limitations with the proposed NeRV. First, to achieve the comparable PSNR and MS-SSIM performances, the training time of our proposed approach is longer than the encoding time of traditional video compression methods. Second, the architecture design of NeRV is still not optimal yet, we believe more exploration on the neural architecture design can achieve higher performances. Finally, more advanced and cutting the edge model compression methods can be applied to NeRV and obtain higher compression ratios.

**Acknowledgement.** This project was partially funded by the DARPA SAIL-ON (W911NF2020009) program, an independent grant from Facebook AI, and Amazon Research Award to AS.

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
