# NeRV: Neural Representations for Videos

**Supplementary Material**

**Hao Chen[1], Bo He[1], Hanyu Wang[1], Yixuan Ren[1], Ser-Nam Lim[2], Abhinav Shrivastava[1]**

[1]University of Maryland, College Park, [2]Facebook AI

{chenh, bohe, hywang66, yxren, abhinav}@umd.edu, sernamlim@fb.com

## A  Appendix

### A.1  NeRV Architecture

We provide the architecture details in Table 1. On a $1920 \times 1080$ video, given the timestamp index $t$, we first apply a 2-layer MLP on the output of positional encoding layer, then we stack 5 NeRV blocks with upscale factors 5, 3, 2, 2, 2 respectively. In UVG experiments on video compression task, we train models with different sizes by changing the value of $C_1, C_2$ to (48,384), (64,512), (128,512), (128,768), (128,1024), (192,1536), and (256,2048).

Table 1: NeRV architecture for $1920 \times 1080$ videos. Change the value of $C_1$ and $C_2$ to get models with different sizes.

| Layer | Modules | Upscale Factor | Output Size & $(C \times H \times W)$ |
|:---:|---|:---:|:---:|
| 0 | Positional Encoding | - | $160 \times 1 \times 1$ |
| 1 | MLP & Reshape | - | $C_1 \times 16 \times 9$ |
| 2 | NeRV block | $5\times$ | $C_2 \times 80 \times 45$ |
| 3 | NeRV block | $3\times$ | $C_2/2 \times 240 \times 135$ |
| 4 | NeRV block | $2\times$ | $C_2/4 \times 480 \times 270$ |
| 5 | NeRV block | $2\times$ | $C_2/8 \times 960 \times 540$ |
| 6 | NeRV block | $2\times$ | $C_2/16 \times 1920 \times 1080$ |
| 7 | Head layer | - | $3 \times 1920 \times 1080$ |

### A.2  Results on MCL-JCL dataset

We provide the experiment results for video compression task on MCL-JCL [1]dataset in Figure 1a and Figure 1b.

### A.3  Implementation Details of Baselines

Following prior works, we used *ffmpeg* [2] to produce the evaluation metrics for H.264 and HEVC.

First, we use the following command to extract frames from original YUV videos, as well as compressed videos to calculate metrics:

```
ffmpeg −i FILE.y4m FILE/f%05d.png
```

Then we use the following commands to compress videos with H.264 or HEVC codec under *medium* settings:

35th Conference on Neural Information Processing Systems (NeurIPS 2021).

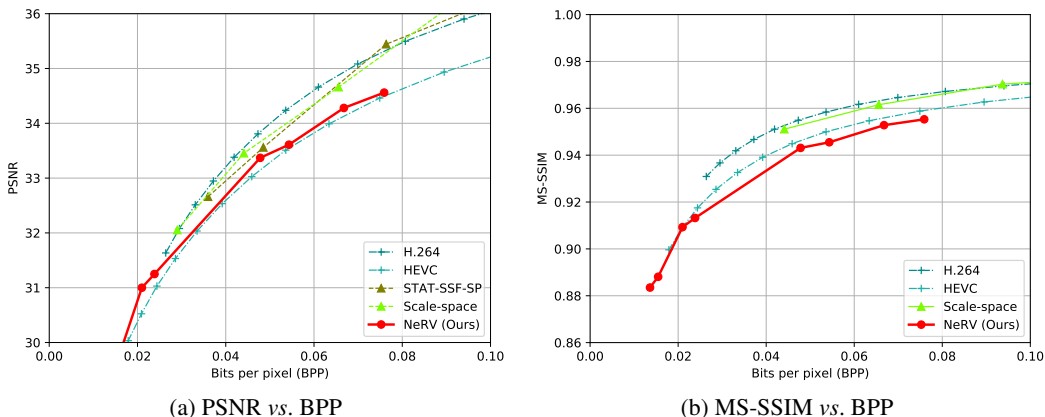

(a) PSNR *vs.* BPP  (b) MS-SSIM *vs.* BPP

Figure 1: Rate distortion plots on the MCL-JCV dataset.

```
ffmpeg -i FILE/f%05d.png -c:v h264 -preset medium \
    -bf 0 -crf CRF FILE.EXT

ffmpeg -i FILE/f%05d.png -c:v hevc -preset medium \
    -x265-params bframes=0 -crf CRF FILE.EXT
```

where FILE is the filename, CRF is the Constant Rate Factor value, and EXT is the video container format extension.

## A.4 Video Temporal Interpolation

We also explore NeRV for video temporal interpolation task. Specifically, we train our model with a subset of frames sampled from one video, and then use the trained model to infer/predict unseen frames given an unseen interpolated frame index. As we show in Fig 2, NeRV can give quite reasonable predictions on the unseen frame, which has good and comparable visual quality compared to the adjacent seen frames.

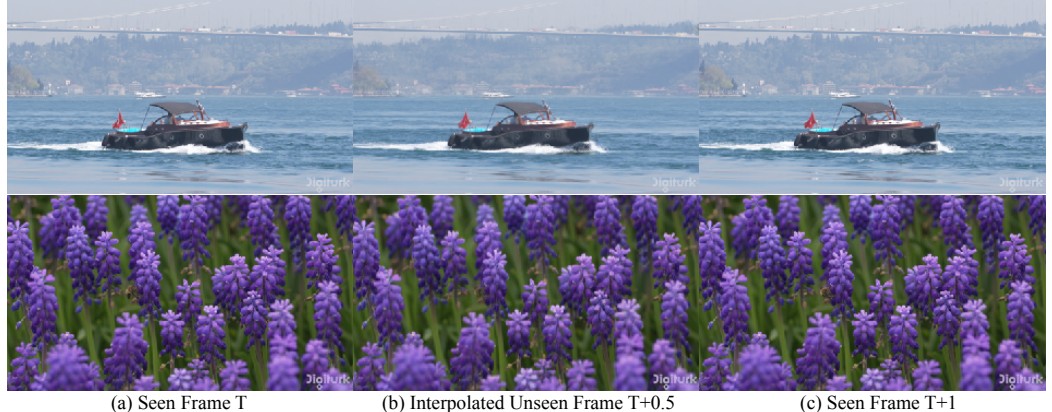

(a) Seen Frame T  (b) Interpolated Unseen Frame T+0.5  (c) Seen Frame T+1

Figure 2: Temporal interpolation results for video with small motion.

## A.5 More Visualizations

We provide denoising results on 'ig buck bunny' video in Figure 3. Given the noisy video as input, NeRV can reconstruct the original video with high fidelity. But it may also over-smooth some high-frequency details in the image and introduce blurry effect.

Besides, we provide more qualitative visualization results in Figure 4 to compare the our NeRV with H.265 for the video compression task. We test a smaller model on "Bosphorus" video, and it also has

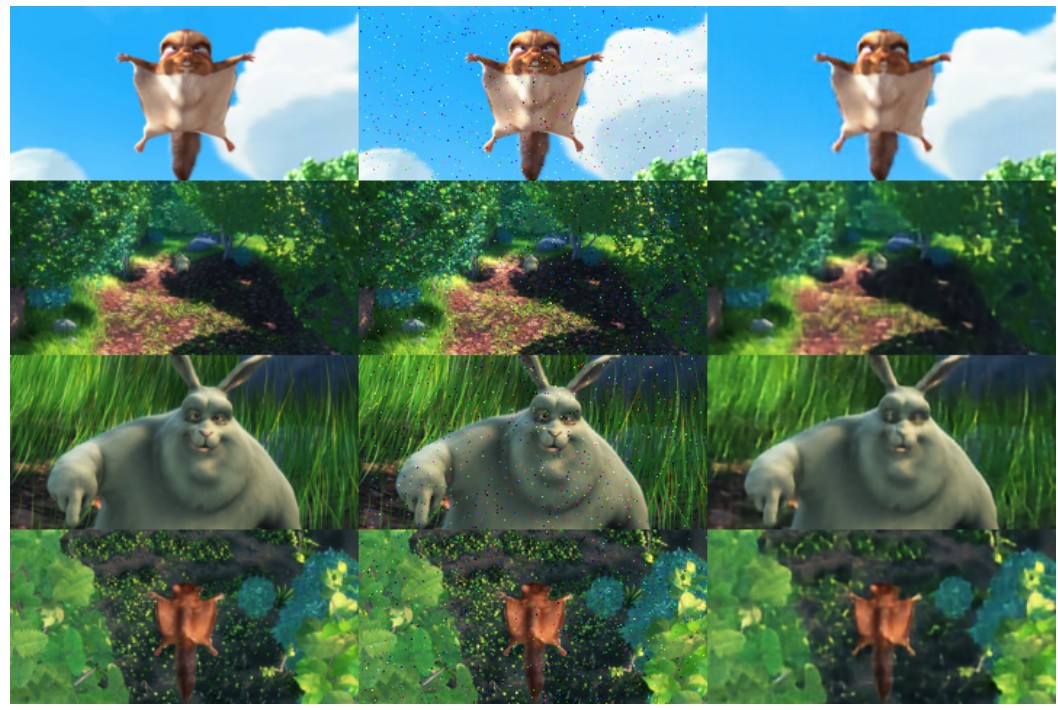

Figure 3: Denoising visualization. **Left**: Ground truth; **Middle**: Noisy input **Right**; NeRV output.

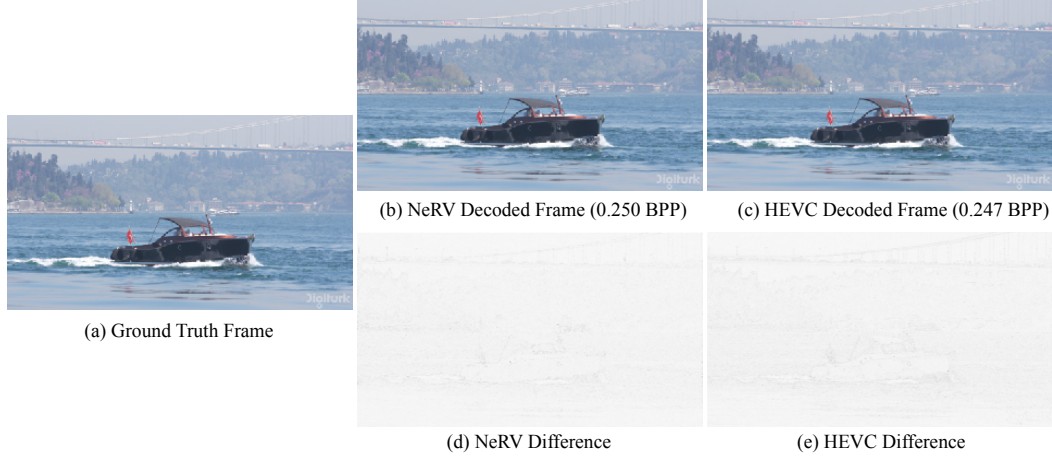

(a) Ground Truth Frame

(b) NeRV Decoded Frame (0.250 BPP)

(c) HEVC Decoded Frame (0.247 BPP)

(d) NeRV Difference

(e) HEVC Difference

Figure 4: Video compression visualization. The difference is calculated by the L1 loss (absolute value, scaled by the same level for the same frame, and the darker the more different). *"Bosphorus"* video in UVG dataset, the residual visualization is much smaller for NeRV.

a better performance compared to H.265 codec with similar BPP. The zoomed areas show that our model produces fewer artifacts and the output is smoother.

**Broader Impact.** As the most popular media format nowadays, videos are generally viewed as frames of sequences. Different from that, our proposed NeRV is a novel way to represent videos as a function of time, parameterized by the neural network, which is more efficient and might be used in many video-related tasks, such as video compression, video denoising and so on. Hopefully, this can potentially save bandwidth, fasten media streaming, which enrich entertainment potentials. Unfortunately, like many advances in deep learning for videos, this approach can be utilized for a variety of purposes beyond our control.