# OpenReview forum: "NeRV: Neural Representations for Videos"
_NeurIPS.cc/2021/Conference — NeurIPS 2021 Poster_

### Official Review · Reviewer_4zsT · 2021-07-11

**Rating:** 5
**Confidence:** 5

**Summary:**

The paper describes a method for representing videos using neural networks. This is inspired by recent advancements in implicit neural representations, which encode various types of signals (audio, images, video, shapes, radiance fields) in the weights of a MLP overfit to the signal. The paper accomplishes this by contributing a new network architecture utilizing convolutional layers which maps each frame index t to the **entire image frame**. Since this network memorizes each frame of the video, compression methods can then be applied on the network instead of the frames themselves to yield a comparable compression versus video quality trade-off. The paper demonstrates that this alternative compression method is comparable to existing standard video compression methods, and that it is robust to noise in the ground truth video.

**Ethical Concerns:**

I do not think there are any ethical issues with this paper.

**Limitations And Societal Impact:**

Yes, the authors have adequately addressed the limitations of their work. They have also addressed the potential negative societal impact of their work.

**Main Review:**

The paper has a number of strengths and weaknesses. In my opinion, the strengths of the paper are:
1. **Clarity**. The paper is written very clearly. The method described is concise, and the paper does a good job of explaining why this sort of representation makes sense for video compression, denoising, and frame interpolation.
2. The quantitative results described are exhaustive, and show that the method does work in memorizing video data using the network architecture described. The rate distortion plots are clear and demonstrate that the method is comparable to existing standard video compression methods, and the tables show that the method has the ability to filter out noise from frames of the video.

In my opinion, the weaknesses are:
1. The related work section misses a few very related works.
    - One major omission is that there is no mention of [SIREN](https://vsitzmann.github.io/siren/) which performs the same experiment of video representation using an implicit neural representation. While I agree that the network architecture used in this case is significantly different (NeRV utilizes convolutional layers to output discrete frames as a function of a continuous frame coordinate, while SIREN also model the pixels in each frame continuously), I think that this work needs to be at the very least mentioned, and ideally compared to.
    - Another major omission is [Deep Image Prior](https://dmitryulyanov.github.io/deep_image_prior), which shows that denoising on images can be done with no supervision, by only training a CNN to represent the image and using early stopping to prevent overfitting. This is highly related to the fact that NeRV is robust to ground truth video noise, and it would be nice to see an ablation, on perhaps a smaller video, to see if given enough training time, the NeRV representation would still learn to memorize the noisy video. This would provide significantly more understanding about *why* this representation is robust to noisy video. A comparison to this method for denoising could also help quantify the difference between NeRV, and the DIP effect.
2. My major concern is with the **significance** of the contribution. I do think the **originality** of the novel network architecture and idea of framing video compression as network compression is worth publishing, in the current state of the paper the results seem not quite complete yet:
    - The limitation of long training time to learn the representation makes this method for video compression not very attractive, especially when considering that quantitatively it only performs on-par with other video compression methods.
    - It’s unclear how the method scales to shorter or longer videos. Would it even be feasible to compress a video which is hours long? No ablations are provided on this aspect, which is very important for evaluating whether this could make an impact in video compression.
    - Getting video frame interpolation for free by using a continuous representation seems appealing, but based on the qualitative results shown in the paper, it does not work quite yet. If this worked well, I think this could significantly improve the quality of the paper since it can also be compared to existing frame interpolation methods.
3. The paper provides limited qualitative comparisons. While seeing the average PSNR or SSIM of the video frames gives some idea of the quality of the estimates, it would make sense to include some sort of qualitative visualization of the result, to make sure there are not artifacts which may induce very small quantitative error, but be very noticeable by human observers. For example, if every other frame was consistently biased away from the GT frame in opposite directions, this could manifest as a very annoying flickering effect in the video which is not caught by the metrics which are used to evaluate quality.

Overall, I like the idea of the paper and think that it has done a good job of making the method simple and clear, and justifying why the modified network architecture and alternative video compression method via representation network compression are valuable and original. However, I think that the significance or impact of the method is limited, since the results are not better than any existing standard compression methods in rate versus distortion, or compression time - the two axes which are most important for video compression. Additionally, the frame interpolation using the continuous representation doesn’t work well enough yet to compare along the frame interpolation axis, and the denoising isn’t studied in depth enough to be sure that NeRV provides novel insight or contribution here. Without these improved results, I am not sure if the ideas of the method alone are ready to be published yet. With improved results, the potential impact of the paper could certainly be improved significantly and make this a much stronger submission in my opinion.

Other comments and questions on the submission:
- typo: On line 189, should reference Table 3 instead of Table 9.
- The section on using a single network to represent multiple videos is not described thoroughly, which makes it hard to understand whether or not some kind of generalization / feature re-use is being done here. Is each video represented by a latent code, which is also input into the network? Or is each video fit in a different part of the input domain? This should be clarified, because I think that this generalization aspect could be one way to improve the rate distortion results, and also provide very interesting insight into feature generalization within implicit neural representations. In my opinion, this explanation and emphasis here could make the submission a lot stronger as well.
- The ablations show that the BatchNorm significantly decreases the quality, but all the experiments are done using a batch size of 1. Does BatchNorm even make sense when the batch size is 1, or does it throw away some information by just normalizing the entire output feature in each layer? Could this be why it does so poorly?
- minor: The references section is inaccurate. For example, [43] was published at NeurIPS 2020, [11] was published at NeurIPS 2019.
- It may be worth to consider changing the name of the method in the title, since NeRV already exists as a publication at CVPR 2021: [NeRV](https://pratulsrinivasan.github.io/nerv/).
- The checklist answers are not completely accurate:
    - 4(b)-(e) are not correct (no new material was included in the submission, and the content of the data used was not discussed). It’s not an issue to answer no or N/A to these questions if no license could be found, for example...
    - 3(d) is not included in the supplementary material

**Changes after Author Responses**:

The response has provided additional comparisons with SIREN architectures, which I think when included in the manuscript should make the contribution of their architecture more impactful. The additional clarifications and comparisons have also cleared up a number of my concerns regarding denoising capabilities, longer videos, and batch size. Thus, I have increased my score. The paper proposes an architecture change, ablates it well, and shows that it gets reasonable results.

However, my concerns remain regarding the limitation of long training time and compression rate compared to existing methods, leading to my borderline assessment of the paper. Without either focusing in depth on *why* the architecture change is an immensely impactful shift in coordinate-based network architectures for compression which will lead to countless future improvements, or providing some kind of utility over baseline methods, I think that the significance of the method is limited.

**Additional Changes**:

The authors have responded to me changing my score by providing more comparisons and an argument related to decoding time being more important than encoding time. I certainly agree with them on this point and appreciate the comparison, and see this as a significant benefit of NeRV. However, shifting the paper to one which tells a story about maximizing a trade-off between compression and decoding time may require a significant amount of additional experiments and ablations (ex: trade-off between PSNR, compression rate, and decoding time). Since there are additional variables here, I don't think I can provide a conditional acceptance in this version of the paper.

As the current version of the paper stands, I think that performance compared to the baselines is too similar to justify the architecture contribution alone. Without either this, or some insight about the architecture change which shows why this is an impactful shift in coordinate-based network architectures for compression which will lead to future improvement, I think the significance of the contribution is not as high as it could be.

I certainly think that there are the makings of a good paper here with some additional experiments and insight, but in its current state with some minor modifications, I will retain my borderline score.


**Time Spent Reviewing:**

3

---

> ### Author Response · Authors · 2021-08-10
> **Addressing concerns by Reviewer 4zsT**
>
> **Q1.1: Comparison with SIREN**
>
> We thank the reviewer for pointing out SIREN and Deep Image Prior as related works to our NeRV.
> * Using spatial index as input, SIREN encodes images/videos through an MLP architecture. Given the large number of pixels in video data, SIREN outputs one pixel value at a time, which makes it much harder to represent video content well. Instead, our NeRV represents the entire video frame as a whole, which simplifies the learning problem and reduces the sample number from T*H*W to T, where T is the frame number, H and W are frame spatial size. Since SIREN only outputs one pixel at a time, its training time is much longer than NeRV due to sampling inefficiency. Compared to NeRV which fully leverages spatial relationships between frame pixels, SIREN can not capture pixels' relations explicitly.
> * To verify the above claim, we conduct the experiment on a small video “Big Buck Bunny sequence” from scikit-video (the same dataset in SIREN paper), which has 132 frames of 720\*1080 resolution. Following the same training schedule from the SIREN paper, we list comparison results in the table below, given a comparable model size, NeRV outperforms SIREN significantly in both PSNR and training time. Note that simply increasing the SIREN model size (3.2M->12M) using a higher hidden dimension and more hidden layers do not help -- it makes the overfitting problem worse due to its dense FC layers. On the contrary, our NeRV always converges to a good solution and achieves better PSNR performance given a bigger model size, as shown below.
>
>
> |       | Model size |  PSNR | Training Time |
> |-------|:----------:|:-----:|:-------------:|
> | SIREN |    3.2M    |  31.3 |      14h      |
> | NeRV  |    3.2M    | 34.06 |     1.5hr     |
> |       |            |       |               |
> | SIREN |     12M    |  28.5 |      20hr     |
> | NeRV  |     12M    | 40.63 |     4.5hr     |
>
>
> * We also conduct experiments by replacing the activation layer with SIREN activations (a periodic sine function) in NeRV architecture, and initialize the MLP parameters as they did. As an extension of Table 6,  we list the results below. We can see that SIREN activations perform much worse than ReLU and Leaky ReLU in our architecture. Our guess is that SIREN activation is only designed for MLP architectures and may not be suitable for MLP+ConvNet.
>
> | Activation Layer |  PSNR | MS-SSIM |
> |------------------|:-----:|:-------:|
> | SIREN activation | 15.74 |  0.4806 |
> | ReLU             | 35.17 |  0.9504 |
> | Leaky ReLU       | 35.59 |  0.9542 |
>
>
>
> **Q1.2: Comparison with Deep Image Prior**
> * Deep Image Prior (DIP) and NeRV are similar in that both train a CNN to represent visual content. However, there are some fundamental differences between them.
> * The effective prior for denoising is different. DIP emphasizes that its image prior is only captured by the network structure because its training data is only a single image. But for NeRV, the training data contains many video frames, which share a lot of visual content. As a result, NeRV’s image prior is captured by both the network structure and the training data statistics.
> * Also, The effectiveness of DIP is highly dependent on Hourglass network architecture, PSNR of the denoising output drops a lot when using ResNet, while NeRV remains stable when using common ResNet architectures. The Hourglass architectures increase the memory footprint, which limits DIP’s application for high-resolution images.
> * Furthermore, we compare the denoising performance of NeRV to DIP. DIP only works if a good early stopping strategy is used to prevent it from overfitting to the noise. Without a noise prior, it has to use fixed iterations settings. By contrast, because NeRV representation is shared by all video frames, which provides a strong regularization to prevent NeRV from overfitting to frame noise. In our experiments, we follow the DIP’s default denoising settings to train the model for 1800 iterations. In addition, we also report 3600 iterations results for a more thorough comparison. The detailed results are listed in the table below. Note that one main purpose of NeRV in our work is video compression, so its network architecture and training design are optimized for video compression instead of denoising like DIP. However, we still see NeRV consistently outperforms DIP in PSNR. Visualization results are also provided in the anonymous link to the AC and can be requested if they permit.
>
> | Frame name                    | Image 1 | Image 2 |
> |-------------------------------|---------|---------|
> | DIP 1800 Iterations (default) | 32.04   | 30.13   |
> | DIP 3600 Iterations           | 30.52   | 32.30   |
> | NeRV                          | 34.90   | 39.27   |
>
>
> **Q2: The limitation of long training time.**
> * Kindly refer to the response to Reviewer RbcY Q3 and Reviewer RbcY Q4.
>
> **Q3: Video concatenation and representing longer videos**
> * First, we would like to clarify ‘video concatenation’. Given 5 videos of length 10s, instead of training one network for each video, we concatenate them into a 50s video first and then represent them with one single network. Given the same frame number, different videos will be identified as different time indexes, [0, 0.2), [0.2, 0.4),, [0.4, 0.6), [0.6,0.8), [0.8,1.0). Through the shared NeRV architecture, every time index will output its corresponding frame. Our hypothesis is that the MLP (feature map output C\*16\*9) will memorize a low-resolution video content through a dense fully-connected layer and be controlled by the time index embedding. Then, the following ConvNet will upsample the low-resolution frame to a high-resolution one, Since the convolution parameters are shared by all spatial locations, during training, video priors can be restored implicitly in the NeRV architecture and parameters.
> * Representing longer or shorter videos is not a problem for NeRV. We list results below that we can fit well on short videos (132 frames) with a small model (~3M/12M) and reach high PSNR. For a long video, a straightforward method to make training feasible is to divide it into multiple short clips and fit one network to one clip, and the distortion rate of the whole video will remain the same with shorter clips. Given enough resources, we can fit one network in such a long video with a reasonable time as stated in Reviewer RbcY Q3.
>
>
>
>
> **Q4: Temporal interpolation**
> * More temporal interpolation visualization results are provided in the anonymized link. Here we emphasize that the temporal interpolation is an interesting and promising application of NeRV rather than the main contribution. The temporal interpolation quality of NeRV is not perfect, but still fairly good, especially given the fact that the design of NeRV is not optimized for temporal interpolation. The temporal interpolation ability is related to generalization. However, as mentioned by Reviewer RbcY, “Since the motivation is to represent a video in full, the network is not trained for generalization, rather it is trained to overfit.” Indeed, this overfitting design intrinsically contradicts the generalization ability. Therefore, to obtain better interpolation performance, we have to trade-off between these two aspects in NeRV’s architecture and training design. We will discuss this in more detail in the manuscript and leave a detailed theoretical analysis of this trade-off for future exploration.
>
> **Q5: Limited impact on video compression, temporal interpolation, and denoising**
> * Kindly see Reviewer mzHf Q3 and mzHf Q4.
>
> **Q6: Training with large batch size**
> * Thank you for pointing out this confusion. We have done experiments with large batch sizes and list them below. Note that the PSNR increases to 33.89, it is still worse than no normalization results. The normalization layer hurts the performance because it contradicts the goal to overfit.
>
> |                 | Batchsize |  PSNR |
> |-----------------|:---------:|:-----:|
> | InstanceNorm    |     1     | 33.83 |
> | BatchNorm       |     4     | 33.94 |
> | None            |     1     | 35.59 |
>
>
> **Q7: Other minor changes**
> * Thank you for pointing these out. We will change the typo in Line 189, the method’s name, the reference format, and the checklist.

---

> > ### Comment · Reviewer_4zsT · 2021-08-26
> > **response**
> >
> > Thank you for the detailed rebuttal. This has addressed many concerns in my review. One follow-up question:
> > > Note that simply increasing the SIREN model size (3.2M->12M) using a higher hidden dimension and more hidden layers do not help -- it makes the overfitting problem worse due to its dense FC layers.
> >
> > I'm not sure that I understand this justification for why the larger SIREN performs worse. The "overfitting problem" is mentioned, but in reality memorizing a video is an overfitting task - the goal is to use the network to represent all of the pixels in the video. So I don't understand why overfitting would be a problem, when the goal is to be able to overfit?

---

> > > ### Author Response · Authors · 2021-08-26
> > > **Clarification**
> > >
> > > * Thank you for your feedback. Here, the “overfitting problem” refers to the task of memorizing a video, the goal for implicit representation. And it is not the reason for SIREN's bad performance.  Usually, a larger model size, e.g., ‘dense FC layers’ in SIREN, means higher representation power and should do better in memorizing, but it can also lead to optimization problems, especially for SIREN with MLP architecture and this is the main reason for its bad performance.
> > > * SIREN’s bad performance comes from the input embedding and network architecture, which makes the optimization much more difficult when using a larger model size. For input embedding, SIREN maps a pixel location pair (t,x,y) to a single pixel RGB value R^3, while NeRV maps the time index t to a whole frame R^(3\*H\*W). Given a 300 frames video with resolution 512\*512, SIREN has T\*H\*W (10^8: 300\*512\*512) pairs to optimize, while we only have T (~300) pairs to optimize. For network architecture, the limitation of MLPs, even by increasing the hidden dimension or adding more hidden layers, could make the optimization more difficult. Just like without convolution and residual connection, simply stacking more FC layers could not tackle the image recognition problem well, even for simple CIFAR recognition. In contrast, our NeRV consists of two parts, MLP (which maps time index to low-resolution feature map, C\*8\*8, etc.) and ConvNet (which maps the low-resolution feature map to the final frame). In all, the optimization complexity (10^8 vs 300 pairs to optimize) and architecture difference (MLP vs MLP+ConvNet) results in the performance gap.
> > > * This is verified by SIREN paper, in Section 3.2 “ *Additionally, we run a simple experiment where we fit a short video with 300 frames and with a resolution of 512×512 pixels using both ReLU and SIREN MLPs. As seen in Figure 2, our approach is successful in representing this video with an average peak signal-to-noise ratio close to 30 dB*”, while in our rebuttal reimplementation, with 3.2M parameters, SIREN PSNR can reach 31.3, but it is still much below our NeRV 34.06. COIN [1] also has such observation (in their Figure 2), with pixel location (x,y) as input and uses MLP as implicit representation for image compression, their PSNR lags far behind other methods due to such simple design and does not reach 30 as well even though they greatly increase the model size.
> > >
> > > |       | Model size |  PSNR | Training Time |
> > > |-------|:----------:|:-----:|:-------------:|
> > > | SIREN |    3.2M    |  31.3 |      14h      |
> > > | NeRV  |    3.2M    | 34.06 |     1.5hr     |
> > > |       |            |       |               |
> > > | SIREN |     12M    |  28.5 |      20hr     |
> > > | NeRV  |     12M    | 40.63 |     4.5hr     |
> > >
> > > [1] Emilien Dupont, Adam Golinski, Milad Alizadeh, Yee Whye Teh, and Arnaud Doucet. "Coin: Compression with implicit neural representations". arXiv preprint arXiv:2103.03123, 2021.

---

> > > ### Author Response · Authors · 2021-09-02
> > > **Addressing significance concern**
> > >
> > > * We do appreciate you upgrading the score. For your **significance** concern, we would like to highlight that decoding time is a more important metric than encoding time, where our method already outperforms other methods significantly (improve FPS **from 1.8 to 12.5**, even surpassing H264’s 9.2). In most real scenarios, one video is encoded only once (e.g., on the server) but it will need to be decoded numerous times (e.g., on client devices). Like movies or videos on YouTube/Vimeo, they need to be encoded *only once*, but will be decoded millions of times to be watched. Therefore, we posit that the decoding time is a much more important metric than the encoding time. We provide these results (at ~0.2 BPP) below and demonstrate the strength of our NeRV representation. Note that our method can be improved further by efficient inference with quantization networks.
> > >
> > > | Methods | Habibian et al. [1] | Wu et al. [2] | Rippel et al. [3] | DVC [4] | Liu et al [5] | H264 | NrRV (Our) |
> > > |:-------:|:-------------------:|:-------------:|:-----------------:|:-------:|:-------------:|:----:|:----------:|
> > > |   FPS   |       10e-3.7       |    1.00E-02   |         1         |   1.8   |       3       |  9.2 |    12.5    |
> > > * For long training time, this can be an inherent attribute for implicit neural representation. However, our method greatly improves the video compression quality and reduces training time compared to the implicit representation baseline. Compared to SIREN, we reduce the training time from **14hr to 1.5 hr** (~10x) and boost PSNR results from 31.3 to 34.06. So if SIREN and other implicit representation papers are considered beneficial after a peer-review process, we urge the reviewer to reconsider the advantages brought by our approach (*much less training time, higher PSNR, denoising, much less decoding time, flexible video size/quality*, etc.), where we show better performance, and therefore believe the community will benefit from it.
> > >
> > > |       | Model size |  PSNR | Training Time |
> > > |-------|:----------:|:-----:|:-------------:|
> > > | SIREN |    3.2M    |  31.3 |      14h      |
> > > | NeRV  |    3.2M    | 34.06 |     1.5hr     |
> > > |       |            |       |               |
> > > | SIREN |     12M    |  28.5 |      20hr     |
> > > | NeRV  |     12M    | 40.63 |     4.5hr     |

---

### Official Review · Reviewer_sHtY · 2021-07-16

**Rating:** 5
**Confidence:** 5

**Summary:**

This paper proposes  a novel neural representation for videos. The key idea is to represent a video as a function conditioned on the time index input. Given a video, it uses a MLP to fit such a function. Two applications are conducted: video compression and video denoising.

**Limitations And Societal Impact:**

The limitation about the time-consuming is very difficult to handle, given the proposed key idea.

**Main Review:**

I cannot see any strengths from this paper. My concerns include:
1) The key idea, to use time index as an condition input is really a common-used strategy in many existing works, like video generation. It lacks insights.
2) Compared with traditional video compression methods, the proposed algorithm requires a video-specific training and a model-pruning post-processing, which needs 3-4 days. This is also stated by the authors in the limitation. In this sense, it lacks applicability.
3) For the experiment on video denoising, training a model to fit a noisy video can finally produce denoised video. This is actually the key insight of "deep image prior" and has no relation to the proposed idea.


**Time Spent Reviewing:**

3

---

> ### Author Response · Authors · 2021-08-10
> **Addressing Reviewer sHtY's concerns**
>
> **Q1: Common-used strategies to use time-index as input.**
> * As we describe in response to Reviewer mzHf Q2, we do not claim the time-index embedding as our contribution. Instead, representing videos as a neural network and verifying this idea with the NeRV framework is the main contribution, which is also noted by Reviewer RbcY and Reviewer 4zsT.
> * “Given a video, it uses a MLP to fit such a function”, we use MLP + ConvNet to generate the video frame instead of a MLP to generate a frame pixel, which is a critical difference from other implicit representation approaches. Using a simple MLP to output a whole video frame introduce a huge parameter burden since its output dimension is huge, 1920\*1080\*3, which makes it impossible to train the network. On the other hand, using a MLP to output one pixel value at a time (SIREN[1]) will have much longer training time and inferior performances. Following the same training schedule from the SIREN paper, we list comparison results in the table below, given a comparable model size, NeRV outperforms SIREN significantly in both PSNR and training time.
> * Also, the simple MLP architecture greatly hinders the representation power of SIREN. Simply increasing the SIREN model size (3.2M->12M) using higher hidden dimensions and more hidden layers does not help -- it makes the overfitting problem worse due to its dense FC layers. On the contrary, our NeRV always converges to a good solution and achieves better PSNR performance given a bigger model size, as shown below.
>
> |       | Model size |  PSNR | Training Time |
> |-------|:----------:|:-----:|:-------------:|
> | SIREN |    3.2M    |  31.3 |      14h      |
> | NeRV  |    3.2M    | 34.06 |     1.5hr     |
> |       |            |       |               |
> | SIREN |     12M    |  28.5 |      20hr     |
> | NeRV  |     12M    | 40.63 |     4.5hr     |
>
>
> **Q2: Applicability considering the training time.**
> * Kindly refer the response to Reviewer RbcY Q3 and Reviewer RbcY Q4
>
>
> **Q3: Discussion with Deep Image Prior paper.**
> * Kindly refer to the response to Reviewer 4zsT Q1.2.
>
> [1] Sitzmann, Vincent, Julien Martel, Alexander Bergman, David Lindell, and Gordon Wetzstein. "Implicit neural representations with periodic activation functions." Advances in Neural Information Processing Systems, 2020.

---

> > ### Comment · Reviewer_sHtY · 2021-09-02
> > **Reply to authors' response**
> >
> > Thanks for the rebuttal, it indeed addressed many of my concerns.
> >
> > However, I still do not agree the argument on training time. For video compression scenario, a long-time training time for each specific video is absolutely not acceptable. The authors argue that this will change along with the development of GPU, but future is unknown. At least, to date, the proposed method has few applicability.
> >
> > So, I change my score to 5 but I still do not think it is a publishing-ready paper.

---

> > > ### Author Response · Authors · 2021-09-02
> > > **Addressing application concern**
> > >
> > > * We do appreciate you upgrading the score. We would like to highlight that decoding time is a more important metric than encoding time, where our method already outperforms other methods significantly (improve FPS **from 1.8 to 12.5**, even surpassing H264’s 9.2). In most real scenarios, one video is encoded only once (e.g., on the server) but it will need to be decoded numerous times (e.g., on client devices). Like movies or videos on YouTube/Vimeo, they need to be encoded *only once*, but will be decoded millions of times to be watched. Therefore, we posit that the decoding time is a much more important metric than the encoding time. We provide these results (at ~0.2 BPP) below and demonstrate the strength of our NeRV representation. Note that our method can be improved further by efficient inference with quantization networks.
> > >
> > > | Methods | Habibian et al. [1] | Wu et al. [2] | Rippel et al. [3] | DVC [4] | Liu et al [5] | H264 | NrRV (Our) |
> > > |:-------:|:-------------------:|:-------------:|:-----------------:|:-------:|:-------------:|:----:|:----------:|
> > > |   FPS   |       10e-3.7       |    1.00E-02   |         1         |   1.8   |       3       |  9.2 |    12.5    |
> > > * For long training time, this can be an inherent attribute for implicit neural representation. However, our method greatly improves the video compression quality and reduces training time compared to the implicit representation baseline. Compared to SIREN, we reduce the training time from **14hr to 1.5 hr** (~10x) and boost PSNR results from 31.3 to 34.06. So if SIREN and other implicit representation papers are considered beneficial after a peer-review process, we urge the reviewer to reconsider the advantages brought by our approach (*much less training time, higher PSNR, video denoising, much less decoding time, flexible video size/quality*, etc.), where we show better performance, and therefore believe the community will benefit from it.
> > >
> > > |       | Model size |  PSNR | Training Time |
> > > |-------|:----------:|:-----:|:-------------:|
> > > | SIREN |    3.2M    |  31.3 |      14h      |
> > > | NeRV  |    3.2M    | 34.06 |     1.5hr     |
> > > |       |            |       |               |
> > > | SIREN |     12M    |  28.5 |      20hr     |
> > > | NeRV  |     12M    | 40.63 |     4.5hr     |

---

### Official Review · Reviewer_mzHf · 2021-07-17

**Rating:** 4
**Confidence:** 4

**Summary:**

Video is encoded as a neural network that takes frame index and outputs the video frame corresponding to the index. Video compression problem is cast as a model compression problem, and the proposed neural representation achieves performance comparable to standard frame-based compression method. The proposed neural representation is demonstrated for temporal interpolation and video denoising tasks.

**Ethics Review Area:**

["I don’t know"]

**Main Review:**

(+) The paper is relatively well organized and easy to follow.
(+) The paper also introduces an interesting idea for representing a video but the benefit/significance is not very clear.

(-) The task mentioned in the paper seems very similar to the copy task introduced in the paper Neural Turing Machine. It would be interesting to compare how the proposed network would perform in the copy task. There is an obvious difference in input and output between the proposed architecture and the NTM. Is it possible to have a neural architecture that is not dedicated to a video but a set of videos would be an interesting question.
Graves, Alex; Wayne, Greg; Danihelka, Ivo (2014). "Neural Turing Machines". arXiv:1410.5401.
(-) The paper does not discuss the philosophy behind architecture. It is difficult for the reader to understand why the network was designed the way it was. For example, there is very little explanation about why the input is encoded as a Fourier feature (is it the only high embedding space to consider) and what the role of pixel shuffle is in the architecture.
(-) Conventional Model compression described in Section 3.1 seems as complex as conventional video compression methods (H264, HEVC). How is the audio of the video encoded?  It would seem that the compression performance will depend on many factors (weight quantization and entropy encoding) which are not fully discussed.
(-) The proposed algorithm does not achieve SOTA and does not present a clear benefit for representing the video in the manner the paper is proposing.
(-) Coordinate-based neural representation is not a novel idea and applying it to video does not seem to give any benefits in terms of performance.
(-) There are various components of the architecture which the authors have compared  (table 4-8) but it is not clear why one would perform better than others. A deeper understanding and intuition would be appreciated.
[1] Readers will appreciate any insights the authors can provide as to how high-frequency variations of the timestamp provides a better result than just the timestamp. Line 128-129 (.. input to high dimensional space seems ... neural network can better fit...) to be suggesting inputting vector representation of scaler values [0, 0.1,0.2,....1] gives better results. The authors do provide additional experimental results using different values of $b$ and $l$ but it does explain why the result would depend on the different representations of deterministic values.
[2] The reader would be interested in the author(s)' elaboration on "burdensome parameters" which is the motivation of upscaling using pixel-shuffle.




**Time Spent Reviewing:**

5

---

> ### Author Response · Authors · 2021-08-10
> **Addressing Reviewer mzHf's concerns**
>
> **Q1: Similar task with Neural Turing Machine**
> * Although the Neural Turing Machine (NTM)’s copy task is very interesting, it is quite different from the proposed NeRV. The copy task in NTM is to remember an input sequence of random binary vectors at test time. Specifically, NTM is trained to be able to memorize different input sequences. During testing, NTM can copy arbitrary sequences, even if they are longer than those seen during training. What NTM learned is the ability to copy, instead of modeling certain content. By contrast, a fully trained NeRV serves as an implicit representation of a video, aiming to faithfully reconstruct the represented video. NeRV does not take any video information as input at testing time. Instead, frame indices are used to query corresponding frames from NeRV.
> * Our NeRV consists of a MLP and ConvNet, where MLP takes frame time index as input and outputs a low-resolution feature map (C\*16\*9), and the ConvNet takes as input the low-resolution feature map and up-samples to the final high-resolution frame (3\*1920\*1080). Since the video frames are consecutive in temporal space, we can use the frame time index as input and reasonable temporal interpolation results verify the assumption. For the copy task of NTM, using addressing index as input may not be the best choice since the neighboring vectors are randomly generated and may have no relationship at all. Such input embedding may not introduce any embedding benefit to remember the memory content. Also, it’s unclear if MLP + ConvNets is a good architecture to copy binary vector sequences.
>
>
> **Q2: Philosophy behind architecture design and input format.**
> * The main idea of NeRV is to represent videos as a neural network, therefore we choose common components to build NeRV architecture and embed time index. This is also verified by Reviewer 4zsT: “The paper is written very clearly. The method described is concise, and the paper does a good job of explaining why this sort of representation makes sense for video compression, denoising, and frame interpolation.” We tried to explain all components of architecture design and input format in the main paper. If the reviewer kindly highlights what was unclear, we will improve the exposition accordingly.
> * For NeRV architecture, we choose the common MLP and ConvNets architectures to output a frame given the frame index. We explore different combinations of activation layers, normalization layers, and upsample methods (Table 4-8 and Table 1 and 2 in the supplementary material).
> * Representing time/spatial information in Fourier features is a common strategy in NLP and computer vision areas, especially for implicit representations [9-18], we have a detailed discussion about this in Sec 2 (pointed out by Reviewer RbcY as well, “Although there have been other techniques that have tried this, they have been clearly mentioned in Section 2”) and Sec 3.1 Line 127-135.
> * PixelShuffle [44] is a common method for upsampling in super-resolution and image generation, which is illustrated in Table 1. We have an ablation study for its comparison with other upsampling methods in Table 4 as well.
>
>
> **Q3: Conventional model compression vs. conventional video compression; encode video audio; full discussion about weight quantization and entropy encoding**
> * We discuss conventional video compression methods in Line 90-107 which have long and complicated pipelines, where each step can be quite complex. Moreover, they treat video as a sequence of separate frames, which can hinder the understanding of video content. As spatial-temporal data, we believe representing videos as a whole (pointed out by Reviewer RbcY as well -- ‘the motivation is to represent a video in full”) can open a new direction in video understanding and processing. We show its potential significance in video compression and denoising (as pointed out by Reviewer 4zsT as well).
> * Audio information is generally not considered in video compression, video interpolation, and video denoising tasks. Therefore, we only focus on the video frames. However, this is an interesting future research direction.
> * We explain weight quantization and entropy encoding in L158-170. We will add more details in the final manuscript.
>
>
> **Q4: SOTA performance and clear benefit**
> * As a novel representation, NeRV, with a straightforward implementation, already achieves a distortion ratio comparable with other SOTA methods in video compression, and outperforms traditional denoising methods for video denoising. We believe many further improvements can be made based on our work (‘great potential impact’ as stated by Reviewer 4zsT). We believe that other researchers in the community will find this direction interesting and urge the reviewer to reconsider their rating.
> * Inference speed of NeRV is another benefit, kindly refer to the response to Reviewer RbcY Q4 for details.

---

### Official Review · Reviewer_RbcY · 2021-07-25

**Rating:** 7
**Confidence:** 5

**Summary:**

The paper proposes to model videos as a function mapping time to image, and uses a neural network to model this function. It trains a neural network on one video or a concatenation of several videos, by overfitting the neural network on the videos. Since the motivation is to represent a video in full, the network is not trained for generalization, rather it is trained to overfit. Then, the paper tries to do video compression by compressing the weights of the neural network. It uses 3 model compression techniques in a cumulative fashion : pruning, weight quantization, entropy encoding. It is found that this gets the compressed representation close to standard video compression techniques in terms of PSNR vs BPP (distortion ratio).

**Limitations And Societal Impact:**

The limitations of the method are clearly described at every stage. The major limitation is that although the performance is comparable to standard techniques, the training time is much higher. Moreover, inference time (time taken to translate from time value to pixel) has not been mentioned. This is an important metric to compare with standard techniques.

Code has been provided to train the model and reproduce results.


**Main Review:**

The paper introduces a simple method for video representation : use the weights of a neural network to represent the video itself. Although there have been other techniques that have tried this, they have been clearly mentioned in Section 2. However, the paper's idea is rather more direct : overfit the weights of a neural network to the video pixels as a function of a Fourier representation of the time index.

This simple idea seems to lead to a representation that is comparable to standard video compression techniques. Ablation studies is provided to explore different aspects of the neural network: 1) PixelShuffle is better for image upsampling, 2) normalization harms the performance (probably because the goal is to overfit), 3) LeakyReLU is better than ReLU, 4) Fourier embedding of time is better than using the time directly. Although each of these ablations studies could be more extensive, the overall network is already comparable to standard techniques in terms of performance, as shown in Figures 5 and 6.

Figure 4 is an insightful plot into how each of the model compression techniques contributes to the method's performance. The conclusions drawn from this plot are reasonable, the significant impact of each compression technique is clearly illustrated.

Apart from video compression, the paper shows the robustness of this method to noisy videos in Table 9. It would be great if it is clarified whether the same noise is added at every batch or whether new noise samples are added per batch. If it is the former, it is indeed impressive that the model is robust to such noise. However, if it is the latter, then it is reasonable that the neural network is learning to ignore the noise for the content.

There is only 1 qualitative example provided in the supplementary. It would be good if more qualitative samples are provided to illustrate the effectiveness of this method.

Another clarification is why the PSNR of videos with random noise (0.3) added is quite high (39.07), even higher than denoised videos.

The paper also makes a case for temporal interpolation. However, only 1 example has been shown (Figure 7). It would be good if more examples are provided.

The paper does mention some of the above as limitations.

Code has been provided to train the model and reproduce results.



Line 189 : Table 3*
Line 220 : Table 3*
Line 233 : bilin*ear
Line 261 : fix*, avoiding*

**Time Spent Reviewing:**

5

---

> ### Author Response · Authors · 2021-08-10
> **Addressing Reviewer RbcY's concerns**
>
>  **Q1: Clarification on video noises and Table 9**
> * Table 9 explores many common noise patterns, where a random i.i.d. noise is added to every pixel with a probability 'p' to generate the final ‘noisy’ video. All methods are performed on these noisy videos.
> * Frame pixel values are normalized between 0 and 1. Random noise (0.3) implies noise ‘r’ is sampled from a uniform distribution (-0.3, 0.3), and the final pixel value is ‘x + r’, where ‘x’ is the original pixel value. For white/black/salt&pepper noise, each frame (of size 1920\*1080) has 1e4 noisy pixels, where white noise means to replace ‘x’ with 1, black noise means replace it with 0, salt&peper noise means randomly choose between the maximum value 1 or minimum value 0. Gaussian noise draws the noise ‘r’ from a    normal distribution N(0, 0.1).
> * For random noise (0.3) and Gaussian noise, since the added noise is relatively small and noisy pixels do not take a large proportion of the frame, the noisy video can still retain high quality and has a high PSNR, 39.07 for random (0.3) and 43.53 for Gaussian. Since the noisy video is of high quality, denoising algorithms will introduce unnecessary blurring and potentially decrease the PSNR value.
> * However, these comparisons are still meaningful, because in practice the noise-prior is unknown and thus denoising algorithms have to be noise-agnostic, which means an effective denoising algorithm needs to be able to be applied to any noises without knowing their specific types and amplitudes, no matter whether they’re significant or not, and thus we evaluate them with many different cases. There can certainly be some cases where applying denoising algorithms actually deteriorates the PSNR results. Table 9 shows that as an added benefit, our NeRV representation can always achieve better results compared to traditional denoising methods for all common noise patterns.
> * We believe the denoising effectiveness comes from two parts: architecture prior and data prior. For architecture prior, the spatial-agnostic property of Convolution operation acts as a strong regularization for natural images. Since noise is added randomly and independent of spatial locations, ConvNets tend to ignore them to work well in other spatial locations. For data prior, the parameter weights are shared across all video frames and act as a strong regularization. Kindly refer to Reviewer 4zsT Q1 for more explanations and experiments. We will add these and improve the exposition in the final manuscript.
>
> **Q2: More visualizations**
> * Since the policy prohibits us from sharing external links directly, we show more visualizations for video compression, temporal interpolation, and video denoising in an anonymous link provided to the AC. If the AC permits it, please request them for the link for the requested visualizations. We also give key highlights from the qualitative results here.
> * For video compression, we display more samples from both datasets, UVG and MCL. We test a larger model on UVG which achieves very good results compared to the ground truth, and outperforms H.265 codec with similar BPP as indicated by the l1 difference (absolute value, scaled by the same level for the same frame, and the darker the more different). We test a smaller model on MCL, and it also has a better performance compared to H.265 codec with similar BPP. The zoomed areas show that our model produces fewer artifacts and the output is smoother.
> * For video temporal interpolation, the interpolated unseen frame has good and comparable visual quality compared to the adjacent seen frames.
> * For video denoising, our method not only outperforms DIP in PSNR and MS-SSIM, but also produces more precise frames in visual quality even quite close to the ground truth. All the visualizations are better zoomed in to view, especially for the details of differences, artifacts and noises.
>
> **Q3: Training time**
> * Even though we mention that the training might take 4-5 days to get comparable PSNR with state-of-the-art methods, getting an *acceptable* video NeRV representation does not take such a long time. As shown in the table below, training for 150 epochs already gives a reasonable NeRV representation and this only takes 5-12hrs.
>
> | Model Size        | 31M              | 48M            | 70.6M            | 99.6M          |
> |-------------------|------------------|----------------|------------------|----------------|
> | PSNR (150 epoch)  | 32.2 (5hr)       | 33.28 (6hr)    | 34.49 (8hr)      | 35.21 (12hr)   |
> | PSNR (2400 epoch) | 32.87 (3.3 days) | 34.04 (4 days) | 35.18 (5.3 days) | 36.37 (8 days) |
>
> * Moreover, we can largely reduce the training time by increasing the batchsize and using parallel multi-GPU training. We list the 150 epoch training time for different batch sizes. We can see that efficient implementation (increasing the batchsize and multi-GPU training) greatly reduces the training time.
>
> | Batchsize | 1     | 2     | 4    | 8     |
> |-------|-------|-------|------|-------|
> | PSNR  | 35.34 | 35.27 | 35.3 | 35.21 |
> | time  | 57hrs | 32hrs | 20hr | 12hr  |
>
> * Finally, better GPUs will also speed up the training time. Compared to Nvidia GeForce RTX 2080 Ti (what we used for experiments), Nvidia Tesla V100 gives ~1.4x speedup and Nvidia Tesla A100 gives ~3x speedup. Given the rapid progress of computation power and efficient deep learning libraries, we believe that the training time will not be a major concern in the near future, and urge the reviewer to consider this.
>
> **Q4: Inference time**
> * We thank the reviewer for the suggestion and totally agree that inference time is an important metric for real-life applications. In most realistic scenarios, one video is encoded once but it will need to be decoded many times. Like a movie, it only needs to be encoded once, but might be watched millions of times. Therefore, the decoding time is a more important metric than the encoding time. We provide these results below and demonstrate the strength of our NeRV representation.
> * All other video compression methods have two types of frames: key & interval frames. Key frame can be reconstructed by its encoded feature only while the interval frame reconstruction is also based on the reconstructed key frames. Since most video frames are interval frames, their decoding needs to be done in a sequential manner after the reconstruction of the respective key frames. On the contrary, our NeRV can output frames at any random time index independently, thus making parallel decoding much simpler. This can be viewed as a distinct advantage over other methods.
> * Below, we compare inference speeds across different methods. Under a similar memory budget (BPP = ~0.2), the decoding speed of NeRV is much faster than other learning-based approaches. Note that H264 is run on CPU, while all other learning-based methods are run on a single GPU. This inference time is a strength of our NeRV representation.
>
> | Methods | Habibian et al. [1] | Wu et al. [2] | Rippel et al. [3] | DVC [4] | Liu et al [5] | H264 | NrRV (Our) |
> |:-------:|:-------------------:|:-------------:|:-----------------:|:-------:|:-------------:|:----:|:----------:|
> |   FPS   |       10e-3.7       |    1.00E-02   |         1         |   1.8   |       3       |  9.2 |    12.5    |
>
> [1]  Amirhossein Habibian, Ties van Rozendaal, Jakub M Tom- czak, and Taco S Cohen. "Video compression with rate-distortion autoencoders". In Proceedings of the IEEE International Conference on Computer Vision, 2019.\
> [2]  Chao-Yuan Wu, Nayan Singhal, and Philipp Kr ̈ahenbu ̈hl. "Video compression through image interpolation". In European Conference on Computer Vision, 2018.\
> [3]  Oren Rippel, Sanjay Nair, Carissa Lew, Steve Branson, Alexander G Anderson, and Lubomir Bourdev. "Learned video compression". arXiv preprint arXiv:1811.06981, 2018.\
> [4] Guo Lu, Wanli Ouyang, Dong Xu, Xiaoyun Zhang, Chun- lei Cai, and Zhiyong Gao. "Dvc: An end-to-end deep video compression framework". In Proceedings of the IEEE Conference on Computer Vision and Pattern Recognition, 2019.\
> [5] Jerry Liu, Shenlong Wang, W. Ma, Meet Shah, Rui Hu, Pranaab Dhawan, and R. Urtasun. "Conditional entropy coding for efficient video compression". In European Conference on Computer Vision, 2020.

---

### Decision · Program_Chairs · 2021-09-27

**Decision:**

Accept (Poster)

**Comment:**

While there was only one reviewer who recommended accepting this work, this positive reviewer made a strong case for why this paper should be accepted.  While there is some concerns about the practical applicability of the proposed method as a real-world compression method, the reviewers have pointed out that a video need only be encoded once in many situations. As such arguments about the amont of time required to encode a video are less strong.  The issue of not handling audio with this work is not relevant. Video compression research papers rarely also address the audio component of a video.

The authors provided a solid rebuttal addressing key points raised by reviewers. In particular, the author response and additional comparisons with SIREN architectures should be integrated into the final manuscript. Reviewers believe that this will help make the contribution of the architecture here more impactful. Additional clarifications and comparisons provided during the discussion phase also cleared up a number of reviewer concerns regarding the denoising capabilities, longer videos, and batch size. In the end the most engaged, but still somewhat negative leaning reviewer asserted that "The paper proposes an architecture change, ablates it well, and shows that it gets reasonable results.".

The positioning of the paper with respect to prior art and contributions was sufficient and by acknowledging the points of the reviewers properly in the final version of this manuscript, I think this work will serve as a good contribution to the conference. The lack of SOTA results is not a reason for rejecting this work. The general idea here is quite interesting and novel enough to warrant acceptance. The idea proposed here is sufficiently interesting and well explored in this work that it would be an interesting contribution to the conference.  This paper is likely to stimulate further interesting discussion at NeurIPS and has potential for further impact.

The AC recommends accepting this paper.